# Comprehensive characterization of plasma cell-free *Echinococcus* spp. DNA in echinococcosis patients using ultra-high-throughput sequencing

Jingkai Ji[1,2,3⊙], Bin Li[4,5⊙], Jingzhong Li[5,6⊙], Wangmu Danzeng[2,7], Jiandong Li[1,2,3], Yanping Zhao[2,3], Gezhen Qiangba[2,3], Qingda Zhang[4], Nibu Renzhen[4], Zhuoga Basang[4], Changlin Jia[4], Quzhen Gongsang[4,6], Jinmin Ma[2], Yicong Wang[8], Fang Chen[8,9], Hongcheng Zhou[2], Huasang[2,7], Jiefang Yin[2,3], Jiandan Xie[1,2,3], Na Pei[2,3], Huimin Cai[2,3,10], Huayan Jiang[2], Huanming Yang[2,11], Jian Wang[2], Asan[2,7], Xiumin Han[10], Junhua Li[2,3,12]*, Weijun Chen[1,2]*, Dong Yang[4,13]*

**1** BGI Education Center, University of Chinese Academy of Sciences, Shenzhen, China, **2** BGI-Shenzhen, Shenzhen, China, **3** Shenzhen Key Laboratory of Unknown Pathogen Identification, Shenzhen, China, **4** Second People's Hospital of Tibet Autonomous Region, Lhasa, China, **5** Tibet Center for Disease Control and Prevention, Lhasa, China, **6** NHC Key Laboratory of Echinococcosis Prevention and Control (Xizang Center for Disease Control and Prevention), Lhasa, China, **7** BGI-Tibet, Lhasa, China, **8** MGI, BGI-Shenzhen, Shenzhen, China, **9** MGI-Wuhan, BGI-Shenzhen, Wuhan, China, **10** People's Hospital of Qinghai, Xining, China, **11** James D. Watson Institute of Genome Sciences, Hangzhou, China, **12** School of Biology and Biological Engineering, South China University of Technology, Guangzhou, China, **13** China-Japan Friendship Hospital, Beijing, China

⊙ These authors contributed equally to this work.
* lijunhua@genomics.cn (JL); chenwj@bgi.com (WC); yangdong1975@sina.com (DY)

**Data Availability Statement:** The data reported in this study are available in the CNGB Nucleotide Sequence Archive (CNSA: https://db.cngb.org/

## Abstract

### Background

Echinococcosis is a life-threatening parasitic disease caused by *Echinococcus* spp. tapeworms with over one million people affected globally at any time. The *Echinococcus* spp. tapeworms in the human body release DNA to the circulatory system, which can be a biomarker for echinococcosis. Cell-free DNA (cfDNA) is widely used in medical research and has been applied in various clinical settings. As for echinococcosis, several PCR-based tests had been trialed to detect cell-free *Echinococcus* spp. DNA in plasma or serum, but the sensitivity was about 20% to 25%. Low sensitivity of PCR-based methods might be related to our limited understanding of the features of cell-free *Echinococcus* spp. DNA in plasma, including its concentration, fragment pattern and release source. In this study, we applied ultra-high-throughput sequencing to comprehensively investigate the characteristics of cell-free *Echinococcus* spp. DNA in plasma of echinococcosis patients.

### Methodology/Principal findings

We collected plasma samples from 23 echinococcosis patients. Total plasma cfDNA was extracted and sequenced with a high-throughput sequencing platform. An average of 282 million read pairs were obtained for each plasma sample. Sequencing data were analyzed

cnsa; accession number CNP0000696). Control datasets of cell lines deep sequencing and simulated data were available with accession number CNP0000857. Control datasets of negative individuals were available with accession number CNP0000868.

**Funding:** This work was supported by the National Key Research and Development Program of China (No.2018YFC1200501), the Infectious Diseases Special Project, Ministry of Health of China (2018ZX10732-401) and the Science, Technology and Innovation Commission of Shenzhen Municipality under grant No. JCYJ20170817145915789. The funders had no role in study design, data collection, and analysis, decision to publish, or preparation of the manuscript.

**Competing interests:** The authors have declared that no competing interests exist.

with bioinformatics workflow combined with *Echinococcus* spp. sequence database. After identification of cell-free *Echinococcus* spp. reads, we found that the cell-free *Echinococcus* spp. reads accounted for 1.8e-5 to 4.0e-9 of the total clean reads. Comparing fragment length distribution of cfDNA between *Echinococcus* spp. and humans showed that cell-free *Echinococcus* spp. DNA of cystic echinococcosis (CE) had a broad length range, while that of alveolar echinococcosis (AE) had an obvious peak at about 135 bp. We found that most of the cell-free *Echinococcus* spp. DNA reads were from the nuclear genome with an even distribution, which might indicate a random release pattern of cell-free *Echinococcus* spp. DNA.

## Conclusions/Significance

With ultra-high-throughput sequencing technology, we analyzed the concentration, fragment length, release source, and other characteristics of cell-free *Echinococcus* spp. DNA in the plasma of echinococcosis patients. A better understanding of the characteristics of cell-free *Echinococcus* spp. DNA in plasma may facilitate their future application as a biomarker for diagnosis.

### Author summary

Echinococcosis is one of the most neglected tropical diseases caused by the metacestodes of *Echinococcus* spp. tapeworms, which affect both humans and livestock. Plasma cell-free DNA (cfDNA) consists of nucleic acid fragments found extracellularly and may contain DNA released from the parasites. Research shows that a variety of parasites can be detected from plasma cfDNA. Cell-free *Echinococcus* spp. DNA in plasma or serum had been tested with PCR-based methods, but these PCR methods had low sensitivity ranged from 20% to 25%. Low sensitivity may be due to our limited understanding of cell-free *Echinococcus* spp. DNA in plasma. Here, we take advantage of high-throughput sequencing to get a comprehensive characterization of cell-free *Echinococcus* spp. DNA. Our results showed that with high-throughput sequencing we could detect cell-free *Echinococcus* spp. DNA in all samples, though at a very low level. Based on the sequencing data, we found that cell-free *Echinococcus* spp. DNA in plasma had a different fragment length distribution to cell-free human DNA, and fragment length distribution of cell-free *Echinococcus* spp. DNA is also different between cystic echinococcosis (CE) and alveolar echinococcosis (AE). The sequencing data can also help trace the release source of cell-free *Echinococcus* spp. DNA from the genome. According to the mapping results of cell-free *Echinococcus* spp. DNA reads, we found that most of them were from the nuclear genome rather than the mitochondrial genome, and their release position showed an even distribution on the genome. These characteristics of cell-free *Echinococcus* spp. DNA in echinococcosis patients' plasma could facilitate their future application in research or clinical settings.

## Introduction

Echinococcosis is a life-threatening zoonosis caused by *Echinococcus* spp. tapeworms with a complex life cycle involving intermediate and definitive hosts. Definitive hosts of *Echinococcus*

spp. tapeworms are mainly carnivores such as dogs, foxes, and wolves, and intermediate hosts are usually ungulates or rodents such as sheep, cattle, and pika [1]. Humans can be accidentally infected and develop echinococcosis [2]. As one of the most neglected diseases, at any given time, echinococcosis is affecting more than one million people globally [3–5]. Among the species in genus *Echinococcus*, there are two most important ones in terms of public health, *E. granulosus* and *E. multilocularis*, responsible for cystic echinococcosis (CE) and alveolar echinococcosis (AE) respectively [6,7]. CE is cosmopolitan, with high endemic areas include western China, Central Asia, eastern Africa, South America, and Mediterranean countries, and AE is mainly in the northern hemisphere [1,7].

The diagnosis of echinococcosis is based on clinical findings, imaging and serological test [7,8]. Imaging includes ultrasound, magnetic resonance imaging, and computed tomography, among which ultrasound is most widely used as the basis for screening and clinical diagnosis [8]. Based on ultrasound observations, the World Health Organization Informal Working Group on Echinococcosis classified CE cysts into six types (cystic lesion or CL, and CE1–5) and AE lesions into different PNM types (Parasite lesion, Neighbor organs, Metastases) [7,8]. Imaging techniques provide the clinician with important clinical information including the location, number, size, and stage of the cysts, which are crucial for the diagnosis of echinococcosis [7,9,10]. However, there are some unsolved issues with imaging techniques, especially the most commonly used ultrasound in diagnosing echinococcosis. The foremost problem is the late diagnosis. As the early phase of infection is generally asymptomatic, patients may remain asymptomatic for years, even permanently. Given the long incubation period (5–20 years), echinococcosis is not easy to be diagnosed in the early stage, and many asymptomatic patients are diagnosed by chance [11–13]. Besides, detecting small cystic lesions is also a challenge in imaging diagnosis of echinococcosis. It is not easy to distinguish echinococcosis cysts from cysts caused by other reasons, such as liver abscesses, Caroli disease, bilomas and cystadenomas [14–17]. The long incubation period and complex clinical manifestation of the disease also makes clinical findings difficult, and patients with symptoms are advised to undergo imaging and serological test immediately, thus clinical finding is of limited added value for diagnosis [7]. A serological test could serve as an auxiliary diagnostic tool, but its limitations include cross-reactivity and incompetence to differentiate present and past infections [18–20]. In consideration of the limitations of the existing diagnosis tools, detecting the cell-free DNA (cfDNA) released by *Echinococcus* spp. tapeworms may serve as a biomarker of the etiological agents [21,22].

CfDNA consists of nucleic acid fragments found extracellularly and mainly exists in the bloodstream, urine and other body fluids [22]. It has been widely used in clinical practice such as non-invasive prenatal testing (NIPT) [23], tumor monitoring [24] and pathogen detection [25]. As for parasite cfDNA, the metabolic activities of the parasites and attacks from the host's immune system may cause the parasites' DNA to be released into the host's circulatory system, and the possible mechanisms can be summarized as active secretion and passive release [22]. Several parasitic diseases have been successfully detected with cfDNA, including *Plasmodium* [26], *Trypanosoma* [27], *Leishmania* [28], *Schistosoma* [29] and *Wuchereria* spp. [30]. Cell-free *Echinococcus* spp. DNA had already been suggested as a biomarker for echinococcosis [21], and its existence in plasma or serum was proven with PCR-based methods [31–33], though with rather low sensitivities (20–25%) [31–33] Low sensitivity prevents further application of using plasma cfDNA in the diagnosis of echinococcosis. The unsatisfactory performance of the previous attempts could be due to three possible reasons. First, it was hypothesized that the cfDNA of the parasite did not enter the blood circulation unless the hydatid cyst(s) ruptured–thus non-existence of the parasite cfDNA in the host blood circulation made this detection method impossible [31]. Secondly, there is cfDNA from the parasite in the blood circulation, but its concentration is too low to be detected by the designed methods. Thirdly, the

understanding of the characteristics of the cfDNA in circulation is limiting the application of cfDNA in detecting the parasitic infection. The better knowledge of cfDNA's characteristics in NIPT has facilitated its improvement from molecular-counting based first-generation testing strategy to global adopted size-based diagnostics [34]. There are studies and reviews on the characteristics of cfDNA in different conditions including cancer, pregnancy, and transplantation [35]. A detailed study on the existence, quantity, and characteristics of cell-free *Echinococcus* spp. DNA in echinococcosis patients' plasma is still missing.

The rapid development of high-throughput sequencing techniques made it feasible to sequence cfDNA in research and medical settings. Compared with target-based PCR methods, sequencing can provide more comprehensive information about cfDNA [25]. High-throughput sequencing of cfDNA has been widely used in tumor and prenatal diagnosis, which provides much more detailed information of cfDNA for clinical practice and research [23,24]. We initiated this study to explore the existence, quantity, and characteristics of cell-free *Echinococcus* spp. DNA in the plasma of echinococcosis patients with high-throughput sequencing. We collected plasma samples from clinically diagnosed echinococcosis patients, produced cfDNA sequencing data with high-throughput sequencing technology, and analyzed the massive data with bioinformatics workflow. The results revealed that high-throughput sequencing of plasma cfDNA could serve as a feasible tool for cell-free *Echinococcus* spp. DNA study and improve our understanding of *Echinococcus* spp. infection in the human body.

## Materials and methods

### Ethics statement

This research was reviewed and approved by the Ethics Committee of Second People's Hospital of Tibet Autonomous Region (SPHTAR-ERC-1), Center for Disease Control and Prevention of Tibet Autonomous Region Institutional Review Board (TCDCP-IRB001) as well as the Institutional Review Board of Beijing Genomics Institute in Shenzhen (BGI-IRB18157-T1). All samples were collected with written informed consent from adult participants, and minors' informed consent was given by their guardians.

### Samples and processing

Blood samples from ultrasound-confirmed echinococcosis patients (N = 23) were collected at diagnosis and before any medical treatment. The patients' gender, age, and clinical classification are shown in Table 1. Type of echinococcosis was classified based on ultrasound observations and classification system of the World Health Organization Informal Working Group on Echinococcosis. Among these patients, 14 subsequently underwent surgical operations to remove the cystic lesions, and 9 received chemotherapy. The only AE case (S1) at the beginning was diagnosed as a cystic lesion with ultrasound examination, and the lesion sample of this case was confirmed with PCR to be *E. multilocularis* infection. All blood samples were collected with Ethylenediaminetetraacetic acid (EDTA) tubes. After collection, plasma samples were stored at 4˚C and centrifuged at 4˚C within four hours. The blood samples were centrifuged at 1600g for 10 min at 4˚C, and the plasma was recentrifuged at 16,000g for 10 min at 4˚C. After centrifugation, plasma samples were immediately stored at −80˚C for further experiments. Samples of the lesion from the 14 surgically treated patients were also collected and stored at −80˚C.

### DNA extraction and high-throughput sequencing

Plasma samples stored at -80˚C were thawed, and cfDNA was immediately extracted from plasma using the cfDNA isolation kit. To yield high-quality cfDNA, two kits were used for

**Table 1. Clinical data and sequencing results of each patient.** A total of 23 echinococcosis patients were involved in the study. Plasma samples were performed with ELISA tests and cell-free DNA sequencing. Lesion samples from surgery patients were performed with PCR tests.

| Patient characteristics | | | Diagnosis characteristics | | | Cell-free DNA sequencing | | | | |
|---|---|---|---|---|---|---|---|---|---|---|
| ID | Gender | Age (Year) | Clinical Type[a] | Lesion samples (PCR) | Plasma samples (ELISA) | Raw Data (PE)[b] | Clean Data (PE)[b] | *Echinococcus* Reads (PE)[b] | *Echinococcus* RPM[c] | *Echinococcus* species |
| S1 | Male | 34 | CL | *E. multilocularis* | positive | 742,927,842 | 615,603,566 | 11140 | 18.096 | *E. multilocularis* |
| S2 | Female | 61 | CE3 | *E. granulosus* | positive | 281,355,103 | 246,691,600 | 2 | 0.008 | *E. granulosus* |
| S3 | Female | 30 | CE1, CE4 | *E. granulosus* | positive | 250,618,825 | 227,517,883 | 3 | 0.013 | *E. granulosus* |
| S4 | Male | 30 | CE3 | *E. granulosus* | positive | 280,431,941 | 249,457,697 | 17 | 0.068 | *E. granulosus* |
| S5 | Male | 40 | CE3 | *E. granulosus* | positive | 279,312,347 | 248,762,189 | 1 | 0.004 | *E. granulosus* |
| S6 | Female | 29 | CE1 | *E. granulosus* | positive | 308,530,827 | 274,958,461 | 4 | 0.015 | *E. granulosus* |
| S7 | Female | 44 | CE2 | *E. granulosus* | positive | 263,251,023 | 236,679,083 | 2 | 0.008 | *E. granulosus* |
| S8 | Male | 29 | CE2 | *E. granulosus* | positive | 360,137,779 | 313,971,836 | 17 | 0.054 | *E. granulosus* |
| S9 | Male | 15 | CL | *E. granulosus* | negative | 351,412,640 | 320,738,789 | 37 | 0.115 | *E. granulosus* |
| S10 | Female | 30 | CE2 | *E. granulosus* | positive | 306,363,351 | 269,830,299 | 13 | 0.048 | *E. granulosus* |
| S11 | Female | 43 | CL | *E. granulosus* | positive | 262,203,537 | 231,673,428 | 173 | 0.747 | *E. granulosus* |
| S12 | Female | 10 | CL | *E. granulosus* | positive | 231,127,477 | 205,492,100 | 1 | 0.005 | *E. granulosus* |
| S13 | Female | 58 | CE3 | *E. granulosus* | positive | 245,838,225 | 219,082,144 | 15 | 0.068 | *E. granulosus* |
| S14 | Female | 36 | CE1 | *E. granulosus* | negative | 256,759,640 | 224,126,469 | 13 | 0.058 | *E. granulosus* |
| N1 | Female | 46 | CE1, CE4 | NA | positive | 244,658,087 | 205,140,224 | 116 | 0.565 | *E. granulosus* |
| N2 | Female | 59 | CE2 | NA | positive | 364,203,245 | 281,310,703 | 129 | 0.459 | *E. granulosus* |
| N3 | Male | 35 | CE2, CE4 | NA | positive | 289,831,896 | 171,116,167 | 367 | 2.145 | *E. granulosus* |
| N4 | Male | 58 | CE5 | NA | negative | 367,706,652 | 213,450,366 | 540 | 2.530 | *E. granulosus* |
| N5 | Male | 14 | CE5 | NA | positive | 248,171,648 | 203,853,580 | 125 | 0.613 | *E. granulosus* |
| N6 | Male | 47 | CE5 | NA | negative | 211,410,500 | 153,887,667 | 234 | 1.521 | *E. granulosus* |
| N7 | Male | 27 | CE1 | NA | negative | 132,535,233 | 114,252,814 | 15 | 0.131 | *E. granulosus* |
| N8 | Female | 49 | CE2 | NA | positive | 117,991,568 | 104,563,714 | 10 | 0.096 | *E. granulosus* |
| N9 | Female | 41 | CE4 | NA | negative | 83,740,668 | 73,423,055 | 18 | 0.245 | *E. granulosus* |

[a] Clinical Type: CL, Cystic lesion. CE1-5, Cystic echinococcosis clinical stage.

[b] PE: Paired-end Reads.

[c] RPM: Read-Pairs Per Million.

cfDNA extraction according to the volume of plasma. Among the 23 plasma samples (S5 Table), 22 samples with volume 0.2 to 0.6 ml were extracted with MagPure Circulating DNA Mini KF Kit (Magen, Guangzhou, China), and one sample (N4) with volume 2.2 ml was extracted with QIAamp Circulating Nucleic Acid Kit (Qiagen, Hilden, Germany). The quantity and quality of cfDNA were assessed with Bioanalyzer 2100 (Agilent Technologies, Santa Clara, USA). The concentration of cfDNA was quantified by Qubit Fluorometer (Invitrogen, Carlsbad, USA) and Qubit dsDNA HS Assay kit (Invitrogen, Carlsbad, USA) following the manufacturer's instructions. As the average fragment length of cfDNA was very short, the usual fragmentation step for library preparation was skipped. The qualified cfDNA was further used to construct sequencing libraries. The final quantified libraries were sequenced on the BGISEQ-500 platform (MGI, Shenzhen, China).

## PCR test of lesion samples

Lesion samples stored at -80˚C were thawed, and DNA was extracted with phenol/chloroform methods. The presence of *Echinococcus* spp. tapeworms DNA in the lesion samples was confirmed with PCR assays which were based on the amplification of a fragment within the

NADH dehydrogenase subunit 1 (ND1) mitochondrial gene [36]. The specific primers and probes with fluorescence can also be used for qualitatively distinguishing *E. granulosus*, *E. omultilocularis*, and *E. shiquicus* [36]. PCR was assayed in a final volume of 30 ul, with 25 ul of master mix and 5 ul of DNA extract, in the ABI 7500 (Applied Biosystems, America) Real-Time PCR System. The thermal cycling condition was: 2 min at 50˚C, 5 min at 95˚C, followed by 40 cycles of 15 sec at 95˚C and 45 sec at 60˚C.

## ELISA test of plasma

Plasma samples of the patients before any medical treatment were assayed with Echinococcosis ELISA IgG kit (Beijing BGI-GBI Biotech, Beijing, China) according to the manual. Briefly, phosphate buffered saline (PBS) diluted plasma samples (1:10) were added to the plates. The plates were incubated for 30 min at 37˚C and then washed five times with the PBS-Tween buffer. Peroxidase-conjugated goat anti-human IgG, diluted 1:2000 in a PBS buffer supplemented with 0.5% Tween-20 and 1.5% BSA, was added to each well and incubated at 37˚C for 30 min. Before the addition of the tetramethyl-benzidine (TMB) substrate, the plates were washed five times with the PBS-Tween buffer. The reaction was stopped by adding 2 mol/L H2SO4. The OD450/630 value was measured by a microtiter plate reader. A positive control sample, a negative control sample, and a blank control sample were included on each plate, with the cut-off value for IgG as 0.18.

## Database construction

Sequences of *Echinococcus* spp. tapeworms were downloaded from the NCBI GenBank database. To reduce sequence contamination and get a high-quality sequence database, all sequences were quality controlled with the following steps. *Echinococcus* spp. sequences from GenBank were chopped into 100 bp short pseudo-reads (step size 50 bp), then mapped to the *Echinococcus* tapeworm common host genome sequences (sheep, cattle, pigs, humans, and mice) with BLASTn [37]. Pseudo-reads with high similarity (identity $\geq$ 97%, coverage $\geq$ 92%, and e-value $\leq$ 1e-5) to the host genome sequences were considered to be from host sequence contamination. These host-contaminated pseudo-reads were located to their original chopped sequence regions, and then the regions were masked with BEDTools [38]. After the above steps, we built a qualified *Echinococcus* tapeworm sequence database.

## Workflow construction

Bioinformatics workflow was constructed to identify *Echinococcus* spp. reads with five main steps (Fig 1). 1) Raw data were first processed with SOAPnuke (v1.5.6) [39] and Fastp (v0.19.5) [40] to remove low-quality reads. 2) Clean data were mapped to *Echinococcus* spp. sequence database with Kraken (v0.10.5) [41], and the candidate *Echinococcus* spp. reads were extracted from mapping results. 3) Remove reads sourced from humans with Snap-aligner (1.0beta.23) [42]. 4) Low-complexity reads were difficult to be classified accurately, thus might introduce false-positive results, and were removed with PRINSEQ (v0.20.4) [43]. 5) Remove reads of other taxa. To remove reads of other microorganisms (such as bacteria, fungus, and viruses) either from plasma or introduced by the experimental process, left candidate reads were separately mapped to the *Echinococcus* database and comprehensive database (NCBI nt) by BLASTn [37]. Reads with poor mapping results (identity < 97%, coverage < 92%, and e-value > 1e-5) to the *Echinococcus* spp. sequences would be removed and reads that had a better mapping result to other species would also be removed.

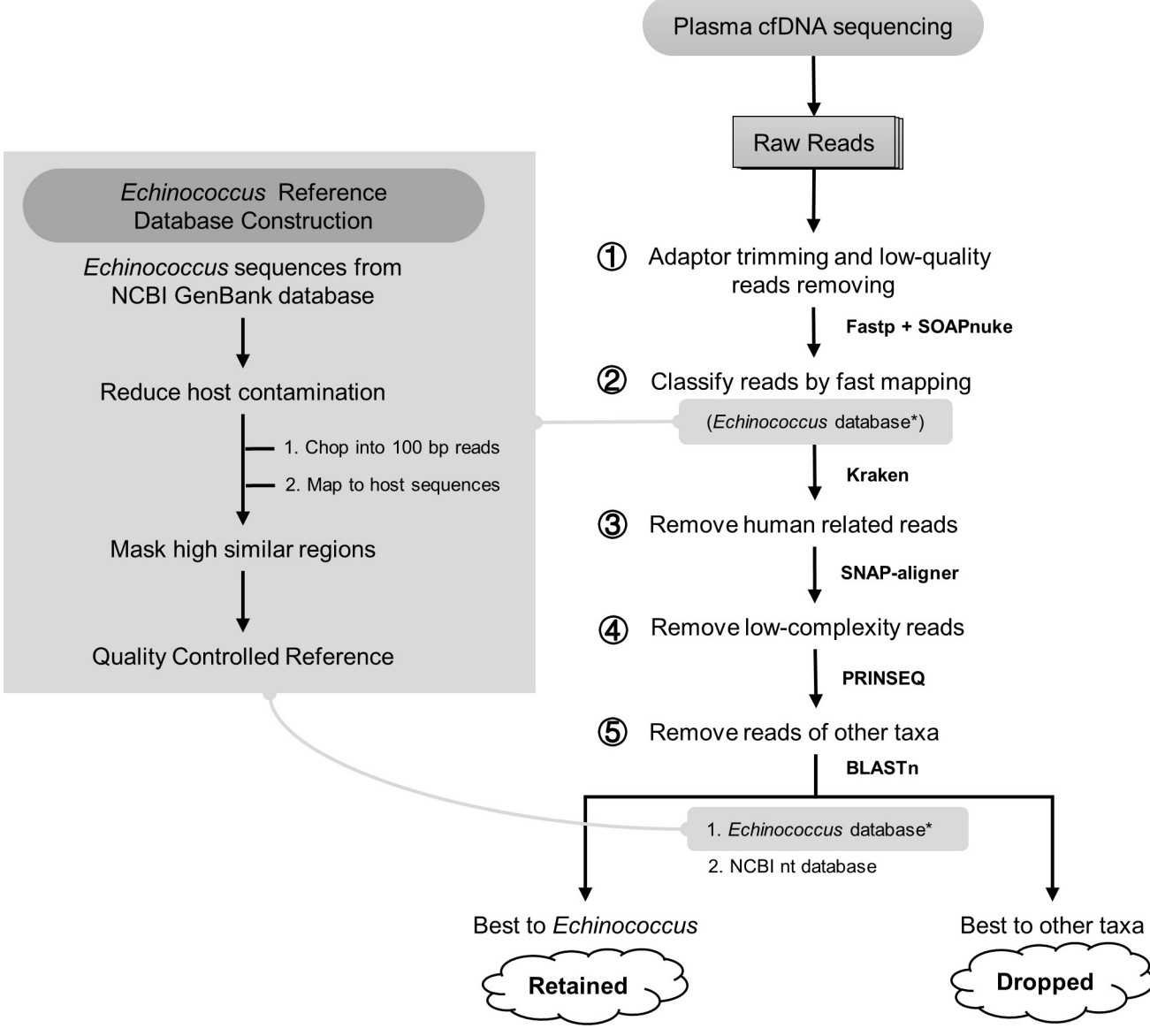

**Fig 1. Reference database construction and analysis workflow.** Construction of *Echinococcus* spp. reference sequence database (left). Analysis workflow of cell-free *Echinococcus* spp. DNA reads identification (right).

## Workflow evaluation

To evaluate the reliability of the workflow, we tested it with three datasets, including simulated data, cell lines deep sequencing data, and cfDNA sequencing data of individuals from non-endemic areas. Simulated data (paired-end 100bp) were produced by wgsim (https://github.com/lh3/wgsim) with human reference and *Echinococcus* spp. sequences. Data of cell lines produced with the same sequencing platform were used as a negative control. CfDNA sequencing data of 107 pregnant women from an ongoing study living in non-endemic areas were also used as a negative control. All three datasets were analyzed with the workflow to evaluate its performance.

### Annotation and fragment length calculation

Identified cell-free *Echinococcus* spp. DNA reads were annotated with information from *Echinococcus* spp. sequence database. The annotation consisted of the release source and the species annotation. The read pairs were labeled according to their best mapping results to the mitochondrial or nuclear sequences which indicated their release source. The species annotation of the sample was determined similarly as the species with the most reads labeled. Based on the samples' species annotation results, *E. granulosus* and *E. multilocularis* [44] were chosen as reference for the cell-free *Echinococcus* spp. DNA features exploration. Since a more complete mitochondrial genome of *E. granulosus* has been published [45], we replaced the mitochondrial sequence of Tsai, *et al.* [44] with the most updated one. Read pairs from *E. granulosus* annotated samples and *E. multilocularis* annotated samples were pooled separately and remapped with BWA (v0.7.16) [46] to their corresponding references to get the mapping positions and fragment length. Based on the mapping results, the insert size was calculated with Picard (http://broadinstitute.github.io/picard). Fragment length distribution figures were produced with R version 3.3.2 (https://www.R-project.org/). Visualization of mapping positions of cell-free *Echinococcus* spp. DNA reads was achieved with Circos [47].

### Analysis of sequencing data volume and positive detection

Based on *Echinococcus* spp. reads proportion, plasma cfDNA concentration, and statistical model, we analyzed the relationship between the amount of sequencing data and positive detection. Sequencing of cfDNA and cell-free *Echinococcus* spp. DNA detection can be regarded as a random sampling process. According to the hypergeometric distribution formula (1), where population size ($N$) = total number of cfDNA fragments, overall target number ($M$) = total number of cell-free *Echinococcus* spp. DNA. The number of draws ($n$) = sequencing reads amount, and the number of observed success ($x$) = detected cell-free *Echinococcus* spp. reads counts. Based on the concentration of cfDNA in the plasma of each sample, we converted the total quality of cfDNA contained in 1ml plasma to base pairs (bp) according to the formula 1pg = 978Mb [48]. According to the literature, the average length of cfDNA is 170 bp [49], and then we estimated total cfDNA fragment counts ($N$) of 1ml plasma. The total number of cell-free *Echinococcus* DNA ($M$) present in 1 ml plasma was estimated based on their proportion detected by sequencing. Then based on the formula (2), we can calculate the probability to get at least one cell-free *Echinococcus* spp. reads detection at a certain amount of sequencing data (S4 Table).

$$P(X = x) = \frac{\binom{M}{x}\binom{N-M}{n-x}}{\binom{N}{n}} \tag{1}$$

$$P(X \geq 1) = 1 - P(X = 0) \tag{2}$$

## Results

### Samples collection and sequencing data production

Blood samples were collected from 23 echinococcosis patients. The average age of these patients (10 males and 13 females) was 38 years (Table 1). Plasma cfDNA was sequenced with the BGISEQ-500 platform and produced a total of 6,480,520,054 paired-end reads with the

amount of data about 1.30 Tb. After quality control, an average of 235,025,384 paired-end clean reads per sample were left.

## Performance evaluation of the analysis workflow

Simulated data, cell lines sequencing data and control human data were used to evaluate the workflow. The simulated data set included 300,000,000 paired-end reads from humans, 1,000 paired-end reads from *Echinococcus* spp. nuclear genome and 100 paired-end reads from *Echinococcus* spp. mitochondrial genome (S1 Table). After analysis with the workflow, 99.5% of the *Echinococcus* spp. nuclear genome reads were identified, 98.0% of the *Echinococcus* spp. mitochondrial genome reads were identified, and no human reads were wrongly identified (S1 Table).

As for the negative controls, DNA of cell lines was sequenced and 2,047,723,953 paired-end clean reads were produced. Evaluation of the cell lines data with the workflow showed that no *Echinococcus* spp. reads were detected. Besides, control data from 107 individuals with a total of 6,838,155,312 paired-end clean reads were used to evaluate the workflow and *Echinococcus* spp. reads were not detected from these data.

## Detection of *Echinococcus* spp. infection

We used cfDNA sequencing and ELISA test to compare their performance in *Echinococcus* spp. infection detection with plasma samples from echinococcosis patients. Sequencing data of plasma cfDNA were analyzed with the analysis workflow. Cell-free *Echinococcus* spp. DNA reads were identified from all the sequencing data (23/23), with an average of 565 read pairs per sample (Table 1). To determine the *Echinococcus* species from cfDNA sequencing data, *Echinococcus* spp. reads were classified with taxonomic information. Species classification results of the identified *Echinococcus* spp. reads showed that 22 samples had most reads annotated to *E. granulosus*, and the remaining sample (S1) had most reads annotated to *E. multilocularis* (S3 Table). In comparison, the ELISA IgG kit identified 17 (73.9%) of the plasma samples of patients (N = 23) as positive. To be specific, out of the 14 surgically confirmed patients, 12 (85.7%) were positive. Out of the 9 non-surgery patients, 5 (55.6%) were positive (Table 1).

Lesion samples from surgery (n = 14) were tested with PCR methods [36] to validate the infection status and identify parasite species. All the 14 lesion samples were PCR positive (Table 1) which confirmed *Echinococcus* spp. tapeworm infection of these patients. According to PCR species differentiation results, 13 lesion samples were identified as *E. granulosus* infection and one lesion sample (S1) as *E. multilocularis* infection. The patient corresponding to S1 should be an AE patient, and other patients were confirmed as CE patients. Species identification results of PCR consisted of sequencing data analysis, which validated the plasma cfDNA sequencing methods.

## Quantification of cell-free *Echinococcus* spp. DNA in plasma samples

To quantify cell-free *Echinococcus* spp. DNA in plasma, we calculated cell-free *Echinococcus* spp. DNA reads proportion in total clean reads of each sample, and the proportion ranged from 1.8e-5 to 4.0e-9 (Fig 2). Given the very low proportion of cell-free *Echinococcus* spp. DNA reads in the sequencing data, we normalized the identified *Echinococcus* spp. reads to total sequencing data with Read-Pairs Per Million (RPM) in order to facilitate comparison between samples. RPM was defined as *Echinococcus* spp. read counts per million sequencing data from one sample. Mean and median RPM of 22 CE patients were 0.433 and 0.082

## *Echinococcus* spp. Reads Proportion

**Fig 2. Cell-free *Echinococcus* spp. DNA reads proportion in total clean reads of the corresponding sample.** A scatter plot shows the detected cell-free *Echinococcus* spp. read pairs proportion ($\log_{10}$) to all clean sequencing read pairs in each sample. The dashed line represents the mean value of 22 *E. granulosus* samples, and the solid line represents their median value. The results showed that the overall concentration of cell-free *Echinococcus* spp. DNA in plasma was at a low level.

respectively (ranging from 0.004 to 2.530) (Table 1), and the RPM of the only one AE sample was 18.096.

Based on the *Echinococcus* spp. DNA reads proportion, we calculated the probability to get at least one cell-free *Echinococcus* spp. read detection at different amounts of sequencing data (S4 Table). The results showed that sequencing with 50 million reads would make 72.73% (16/ 22) of CE samples with over 80% probability to get positive results, sequencing with 200 million reads would make 90.91% (20/22) of CE samples with over 80% probability to get positive results, and sequencing with 400 million reads would make all 22 CE samples with over 80% probability to get positive results.

## Release source of cell-free *Echinococcus* spp. DNA

By reads mapping to the reference genomes, we traced cell-free *Echinococcus* spp. DNA to their genome release source. The analysis showed that most reads were from the nuclear genome, and only a small proportion was released from the mitochondrial genome. A small amount of mitochondrial sourced reads was identified in only 7 CE samples (7/22) and the average proportion was 2.08% (ranging from 0.74% to 7.69%) (S2 Table). To calculate reads per genome size, we normalized the reads counts by the genome size of nuclear and mitochondrial (S2 Table). For the seven CE samples detected with mitochondrial reads, reads per genome size of mitochondrial were from 5.66e-5 to 3.39e-4, and reads per genome size of nuclear were from 1.05e-7 to 4.68e-6. The reads per genome size value of mitochondria are all higher than that of nuclear, and the value of mitochondria was between 48.35 and 539.96

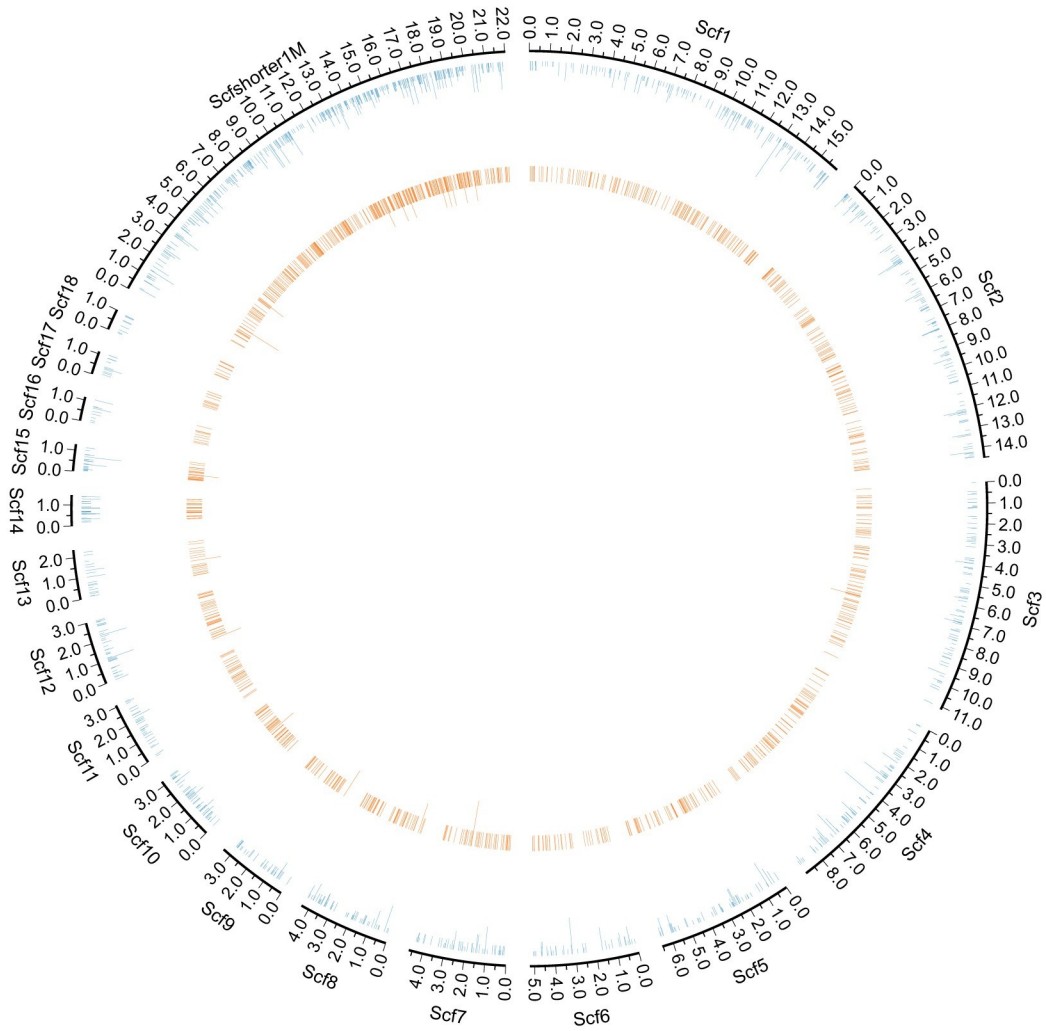

**Fig 3. The distribution of cell-free *E. granulosus* DNA reads on the nuclear genome.** The circulation genome visualization showed the *E. granulosus* reads mapping position on the nuclear genome (outermost blue circle). Eighteen scaffolds longer than 1Mb were displayed in the separate fragments (Scf1-Scf18). Scaffolds shorter than 1Mb were concatenated to display (Scfshort1M). The inner orange circle represents the count of patients with reads detected in the region. Circle figures of the *E. granulosus* mitochondrial genome were put in the supplementary materials (S1 Fig).

(median 75.78) times that of nuclear in the seven CE samples. For the AE sample, the mitochondrial sourced reads proportion was 0.19%, and reads per genome size of mitochondrial and nuclear were 1.54e-3 and 9.67e-5.

Based on the reads mapping, we further traced the release positions of cell-free *Echinococcus* spp. DNA from the genome. Given the low proportion of cell-free *Echinococcus* spp. DNA reads from the 22 CE samples, we pooled their reads and got a total of 1,852 read pairs. The number of cell-free *Echinococcus* spp. DNA read pairs of the AE sample was 11,140. These reads were mapped to the reference genomes of *E. granulosus* and *E. multilocularis* separately. The reads coverage of *E. granulosus* was 213,587 bp, which accounted for about 0.19% of the whole reference genome. The reads coverage of *E. multilocularis* was 1,232,072 bp, which accounted for about 1.07% of the whole reference genome. Mapping positions of cell-free *Echinococcus* spp. DNA reads showed that they appeared to be evenly distributed across the genomes (Fig 3, S1–S5 Figs)

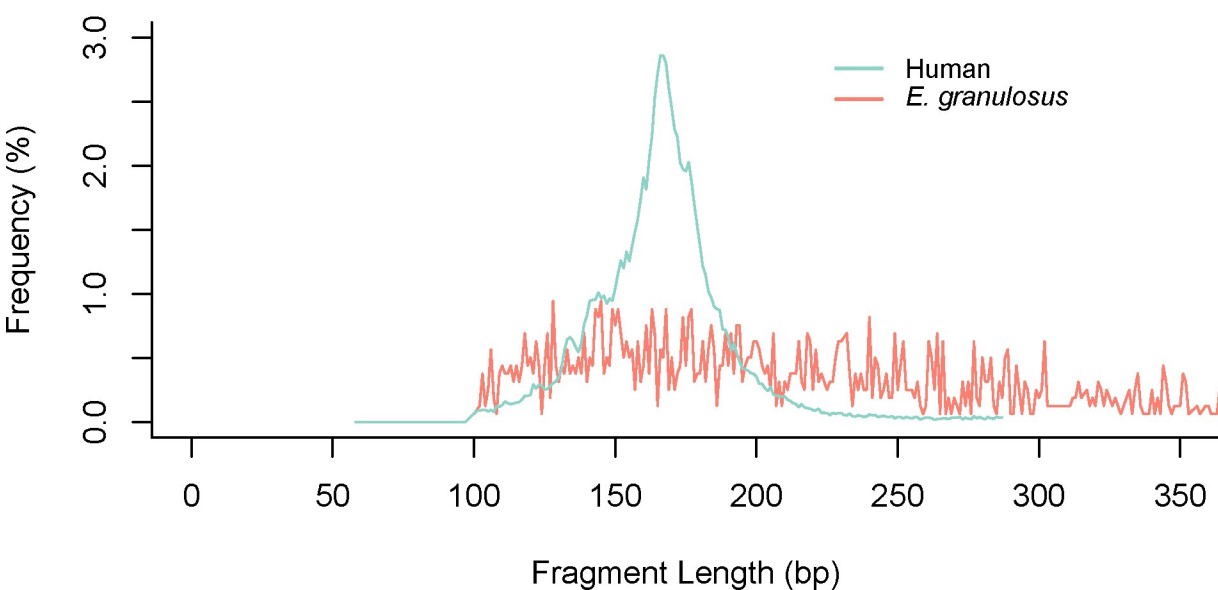

**Fig 4. Fragment length distribution of cell-free *E. granulosus* DNA.** The fragment length of cfDNA was calculated by the insert size of read pairs. The fragment length of cell-free *E. granulosus* DNA had a broad range than human cfDNA.

To analyze the distribution of identified reads in the genome, we calculated the coverage of these reads. Total coverage of 22 CE samples' reads were 213,587 bp. To analyze the overrepresented regions, we calculated the sample counts of the mapped regions. Most of the regions (211,342 bp, 98.95%) were covered with only one sample, less than 1% mapped length (1,983 bp, 0.93%) were covered with two samples, and very small region (262 bp, 0.12%) were covered with three samples. In order to intuitively compare the coverage between different samples, we plot the coverage of the samples detected with more than 100 read pairs (S2 Fig and S3 Fig).

## Fragment length of cell-free *Echinococcus* spp. DNA

According to reads mapping to *Echinococcus* spp. genome references, fragment length of cell-free *Echinococcus* spp. DNA was inferred from the insert size of the read pairs. While human cfDNA showed an obvious peak at around 166 bp, cell-free *Echinococcus* spp. DNA fragment length distribution showed a different pattern. Cell-free *Echinococcus* spp. DNA fragment of CE showed a longer length range without an obvious peak (Fig 4). Cell-free *Echinococcus* spp. DNA fragment of the AE sample showed a more regular distribution pattern with an obvious peak at about 135 bp, which was shorter than human cfDNA (S6 Fig).

## Discussion

With ultra-high-throughput sequencing technology, using plasma samples from clinically diagnosed echinococcosis patients, we identified the existence of cell-free *Echinococcus spp.* DNA in plasma, quantified the amount per sample, confirmed its low concentration and described its characteristics. The results revealed that high-throughput sequencing of plasma cfDNA could serve as a useful tool for cell-free *Echinococcus* spp. DNA studies and improve our understanding of *Echinococcus* spp. infection in the human body. Plasma cfDNA has shown its usefulness in NIPT [23], tumor monitoring [24], and pathogens detection [25].

Several attempts were made using cfDNA in *Echinococcus* spp. detection from plasma or serum with PCR-based methods, but their overall sensitivity was only 20% to 25% [31–33]. The low sensitivity could be due to non-existence, or low concentration of cfDNA of the parasite in the circulation, which showed our limited understanding of the cfDNA of *Echinococcus spp.* tapeworms. As one of the most neglected tropical diseases and zoonosis, echinococcosis poses serious public health threats to endemic areas. Given the increase of global trade, tourism, and immigration, people of non-endemic regions could also be diagnosed with echinococcosis [50–52]. Effective detection and diagnosis methods are the premises of controlling echinococcosis, and cfDNA could be a promising tool for clinical diagnosis. We are the first using high-throughput sequencing technology to evaluate the existence, quantity, and characteristics of cell-free *Echinococcus* spp. DNA in plasma of echinococcosis patients.

The existence of *E. granulosus* DNA in the blood circulation of the echinococcosis patients was questioned by Chaya *et al.* who believed that the cfDNA of the parasite would only enter the blood circulation when the hydatid cyst(s) ruptured [31]. Baraquin *et al.* confirmed the existence of cfDNA of *E. multilocularis* in AE patients and used the very low concentration of cfDNA to explain the low sensitivity of their PCR test [32]. Low concentration of target DNA in plasma is a common situation for cfDNA studies. Cell-free fetal DNA in maternal plasma cfDNA accounts for about 10% to 15% [53,54], and circulating-tumor DNA comprises about 0.01% to 10% or more in cancer patients plasma cfDNA [55,56]. Based on high-throughput sequencing data and bioinformatics workflow, we identified the cell-free *Echinococcus* spp. DNA reads from sequencing data of all the samples. Compared with cell-free fetal DNA and circulating-tumor DNA, cell-free *Echinococcus* spp. DNA in plasma is extremely low, whose proportion ranged 1.8e-5 to 4.0e-9 in these samples (Fig 2). Indeed, this low concentration may explain the low sensitivity of the PCR-based methods [31–33]. Besides the low concentration, we identified the difference between the cell-free *Echinococcus* spp. DNA from CE and AE samples. The AE sample had much more cfDNA identified than the CE samples, which could be due to the different developmental mechanisms of metacestode in the human body. Compared with *E. granulosus*, the metacestode of *E. multilocularis* is an infiltrating lesion composed of aggregated microvesicles, necrosis cells, and fibrosis cells, which have no clear edge to the host tissues [1] and relatively high concentration of cell-free *Echinococcus* spp. DNA in the AE sample could be due to the mixture of necrotic parasite tissue and actively proliferating tissues. This is in line with the previous finding that the sensitivity of PCR-based methods in AE samples was higher than in CE samples [31–33]. As we only collected one AE sample, it needed further verification with more samples. Plasma samples were also tested with ELISA assays to detect the antibody, and the positive results were found in 16 out of 22 CE patients (Table 1). Serological tests may be influenced by lots of factors, and difficult to standardize [18–20]. In contrast, DNA detection is a more direct and objective biomarker.

Low concentration is the major challenge to apply cell-free *Echinococcus* spp. DNA testing in routine clinical settings. To estimate the minimal number of reads needed to get cell-free *Echinococcus* spp. DNA, we treated sequencing as a random sampling process, and the number of sequencing reads regarded as sampling times. We estimated total cfDNA fragment counts and *Echinococcus* spp. fragment counts according to cfDNA concentration and existed detection results. By using hypergeometric distribution, we calculated the probability of each sample to get cell-free *Echinococcus* spp. DNA. detection with different sequencing amounts. We found that sequencing with 50 million reads would make 72.73% (16/22) of CE samples with over 80% probability to get positive results, while sequencing with 400 million reads would make all 22 CE samples identified with over 80% probability to get positive results (S4 Table). The lower the concentration, the harder it is to be detected, and increasing the amount of sequencing can increase the chance of positive detection. The concentration may vary greatly

between individuals. Just like the cell-free DNA of fetus in maternal plasma, which are influenced by gestational age, maternal BMI, fetal aneuploidy status and other factors [57]. The concentration of cell-free *Echinococcus* spp. DNA might also be affected by many factors, such as disease status, parasite species, lesion size, and position, which need more comprehensive samples to explore its association with different patterns.

Cell-free DNA *Echinococcus* spp. DNA in plasma could not only detect the etiology of the patients' infection but also facilitate the species identification. Traditional species identification of *Echinococcus* spp. in echinococcosis patients is always invasive, which relies on the product of surgery or puncture. Surgery is only recommended for part of echinococcosis patients, puncture can assist to get specimens for confirming etiology. However, while puncture is of high diagnostic value and safe in most AE patients [58], it is not recommended for some CE patients, especially for CE4, CE5 and lung cysts, which may pose the risks of allergic reactions and anaphylaxis [1,8]. In this study, species annotation of cell-free *Echinococcus* spp. DNA was analyzed according to reads mapping results. Given the genome sequence similarity between *Echinococcus* species and limited reference sequences available, part of cell-free *Echinococcus* spp. DNA reads may be classified into closely related species of genus *Echinococcus*, but the majority of the reads should be classified correctly. Consistency of species classification between cell-free *Echinococcus* spp. DNA and lesion samples' PCR results proved their accuracy in species annotation. This cfDNA sequencing-based taxonomy annotation method may provide an innovative non-invasive alternative to obtain more detailed etiology information. Species identification of echinococcosis patients could provide more valuable information for guiding clinical management and research such as molecular epidemiology [59].

Tracing cell-free *Echinococcus* spp. DNA release sources could provide more background information. Based on cell-free *Echinococcus* spp. DNA reads mapping, we further analyzed their genome release source. Sequence origin analysis showed that much more cell-free *Echinococcus* spp. DNA was released from the nuclear genome than the mitochondrial genome. This phenomenon may be due to the fact that the genome size of nuclear is much larger than mitochondria. The overall low proportion of mitochondrial-derived cell-free *Echinococcus* spp. DNA in plasma may also partially explain the low positive rate of mitochondrial gene based PCR [31–33]. However, reads per genome size of mitochondria were about 75.78 times larger than that of nuclear, which could be due to the multi copies of mitochondria [44]. The position distribution of cell-free *Echinococcus* spp. DNA on the genome were analyzed with reads mapping to *E. granulosus* and *E. multilocularis* genome references. We found that the release positions of cell-free *Echinococcus* spp. DNA were nearly evenly distributed on the genome. It looks like there are some hotspots of cell-free *Echinococcus* spp. DNA release on the genome, but these spots are more gathered on the short and not well-assembled regions of the available genome references. With higher quality references in the future, the distribution of cell-free *Echinococcus* spp. DNA on the genome could be more evenly distributed.

Size characteristics of cfDNA is an important biological property [35]. To have a deep understanding of cell-free *Echinococcus* spp. DNA, we analyzed its fragment size with sequencing data. Literature shows that cfDNA could have different size pattern according to research settings [35]. Fetal cfDNA in maternal plasma has a shorter fragment size distribution compared with maternal cfDNA [60]. In certain types of cancer patients, tumor sourced cfDNA is concentrated in short fragments [61]. Fragment size analysis of cell-free *Echinococcus* spp. DNA in our study showed that they had a different length distribution to human-sourced cfDNA. We found that cell-free *Echinococcus* spp. DNA of CE had a broad length range (Fig 4), but that of AE had an obvious peak at about 135 bp (S6 Fig). The size profile of cfDNA is relevant to their release mechanism such as apoptosis, necrosis and actively release [62,63]. Quite different fragment size features of cell-free *Echinococcus* spp. DNA in CE and AE could

be related to their developmental mechanism of metacestode in the human body. Tumor like AE lesions may give some explanation to its overall short fragment length, and similar phenomenon of tumor-derived DNA in plasma of hepatocellular carcinoma patients was also observed [64]. As there was only one accidental AE sample, this phenomenon needs more research to validate. Though the exact release mechanism of cfDNA is still unclear, it doesn't affect the application of size properties in diagnostics [35]. As for cell-free *Echinococcus* spp. DNA, their fragment size features may facilitate their detection in future studies.

The cfDNA sequencing-based method relies on high quality and comprehensive database, but existing genome references of *Echinococcus* spp. are limited, and only several genome references are available [44,65,66] whose quality is far from perfect. More importantly, sequence contamination is a serious problem for cell-free *Echinococcus* spp. DNA detection and contaminated sequence database might introduce false-positive results. Since the *Echinococcus* spp. tapeworm samples are always separated from host tissue [44,65,66], it is not easy to remove the contamination of host thoroughly by experimental processing. In the process of genome constructing, some host sequences may mix into the parasite sequence, which is a common problem for genomes construction [67]. It is essential to qualify the genome sequence with bioinformatics methods after downloading from the public database, instead of using it directly [67]. In our study, we filtered the *Echinococcus* spp. sequence database with their common host genomes such as sheep, humans, and mice, and evaluated the workflow with simulation data, cell line data, and negative control data, which all showed that qualified database introduced no false-positive results.

High-throughput sequencing facilitated identifying, quantifying and analyzing the characteristics of cell-free *Echinococcus* spp. DNA in human plasma. These comprehensive characteristics could help the application of cell-free *Echinococcus* spp. DNA in the future diagnosis of echinococcosis. However, for the very low concentration of cell-free *Echinococcus* spp. DNA, their even distribution on the genome, and the high sequencing depth and cost, the method requires further optimization. To increase the application of cell-free *Echinococcus* spp. DNA, we could think of some areas to be explored in the future study, for example, capturing cell-free *Echinococcus* spp. DNA with probes covered the whole genome and enriching the concentration of cell-free *Echinococcus* spp. DNA by host sequence removal. As for clinical application scenarios, massive sequencing of plasma cfDNA to detect cell-free *Echinococcus* spp. DNA may not be suitable for routine clinical examination yet, but it could be used for differential diagnosis, in which existing clinically methods cannot give clear conclusions. For example, the CL patients can be further diagnosed with plasma cfDNA sequencing and avoid the risk of invasive diagnosis.

## Supporting information

**S1 Table. Evaluation of analysis workflow with simulation data.** Simulation data showed that no human reads appeared in the results, and most *Echinococcus* spp. reads were identified by the analysis workflow. Counts in the table were read pairs.
(XLSX)

**S2 Table. Release source of identified *Echinococcus* spp. reads.** Most of the identified *Echinococcus* spp. reads were released from the nuclear genome. Only eight samples were identified with mitochondrial reads.
(XLSX)

**S3 Table. Species classification with cfDNA reads mapping.** The table showed the species classification results from each sample with cfDNA sequencing read pairs. The sample was

classified to the species with most read pairs mapping.
(XLSX)

**S4 Table. Amount of sequencing data and reads detection.** Statistical analysis with hypergeometric distribution to estimate the probability to get positive results with different sequencing amount.
(XLSX)

**S5 Table. Plasma volume and DNA extraction methods.** The volume of plasma and kit used for each sample.
(XLSX)

**S1 Fig. Circle figure of *E. granulosus* samples based on the mitochondrial genome.** The circulation genome visualization showed the *E. granulosus* reads mapping position (outermost blue circle). The inner orange circle represents the count of patients with reads detected in the region.
(TIFF)

**S2 Fig. Circle figure of multiple *E. granulosus* samples based on the nuclear genome.** Seven *E. granulosus* samples detected with more than 100 *Echinococcus* spp. read pairs were displayed based on the nuclear genome. Green and red circles indicate different samples.
(TIF)

**S3 Fig. Circle figure of multiple *E. granulosus* samples based on the mitochondrial genome.** Seven *E. granulosus* samples detected with more than 100 *Echinococcus* spp. read pairs were displayed based on the mitochondrial genome. Green and red circles indicate different samples.
(TIF)

**S4 Fig. Circle figure of the *E. multilocularis* sample based on the nuclear genome.** The circulation genome visualization showed the *E. multilocularis* reads mapping position (outermost blue circle). Ten scaffolds longer than 1Mb were displayed in the separate fragment (Scf1-Scf10). Scaffolds shorter than 1Mb were concatenated to display (Scfshort1M).
(TIF)

**S5 Fig. Circle figure of the *E. multilocularis* sample based on the mitochondrial genome.** The circulation genome visualization showed the *E. multilocularis* reads mapping position (outermost blue circle).
(TIF)

**S6 Fig. Fragment length distribution of the *E. multilocularis* sample.** Overall fragment length distribution of *E. multilocularis* cfDNA was shorter than that of humans.
(TIFF)

## Acknowledgments

The authors would like to thank China National GeneBank for sequencing the samples of this study. We would also like to thank Yue Zhao and Ying Yi for their previous analysis work on the project.

## Author Contributions

**Conceptualization:** Bin Li, Jingzhong Li, Jinmin Ma,  Asan, Xiumin Han, Weijun Chen, Dong Yang.

**Data curation:** Wangmu Danzeng, Yanping Zhao, Yicong Wang, Fang Chen.

**Formal analysis:** Jingkai Ji, Jiandan Xie, Huimin Cai, Junhua Li.

**Funding acquisition:** Bin Li, Jingzhong Li.

**Investigation:** Jingkai Ji, Qingda Zhang, Nibu Renzhen, Zhuoga Basang, Changlin Jia, Quzhen Gongsang, Jinmin Ma, Yicong Wang, Fang Chen,  Huasang, Xiumin Han.

**Methodology:** Jingkai Ji, Jiandong Li, Gezhen Qiangba, Na Pei, Junhua Li.

**Project administration:** Jiefang Yin, Huayan Jiang, Weijun Chen, Dong Yang.

**Resources:** Jingzhong Li, Wangmu Danzeng, Nibu Renzhen, Zhuoga Basang, Changlin Jia, Quzhen Gongsang, Hongcheng Zhou,  Huasang.

**Software:** Jingkai Ji, Jiandong Li, Jiandan Xie, Huimin Cai, Junhua Li.

**Supervision:** Qingda Zhang, Jiefang Yin, Huanming Yang, Jian Wang,  Asan, Weijun Chen, Dong Yang.

**Validation:** Jiandong Li, Gezhen Qiangba, Na Pei.

**Visualization:** Jingkai Ji, Jiandong Li, Gezhen Qiangba, Huimin Cai.

**Writing – original draft:** Jingkai Ji.

**Writing – review & editing:** Bin Li, Jingzhong Li, Yanping Zhao, Junhua Li, Weijun Chen, Dong Yang.

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
