## [Decision Letter · Decision Letter 0]

26 Nov 2019

Dear Mr Ji:

Thank you very much for submitting your manuscript "Comprehensive characterization of cell-free Echinococcus spp. DNA in echinococcosis patients’ plasma using extremely high throughput sequencing." (#PNTD-D-19-01705) for review by PLOS Neglected Tropical Diseases. Your manuscript was fully evaluated at the editorial level and by independent peer reviewers. The reviewers appreciated the attention to an important problem, but raised some substantial concerns about the manuscript as it currently stands. These issues must be addressed before we would be willing to consider a revised version of your study. We cannot, of course, promise publication at that time.

We therefore ask you to modify the manuscript according to the review recommendations before we can consider your manuscript for acceptance. Your revisions should address the specific points made by each reviewer. 

When you are ready to resubmit, please be prepared to upload the following:

(1) A letter containing a detailed list of your responses to the review comments and a description of the changes you have made in the manuscript.

(2) Two versions of the manuscript: one with either highlights or tracked changes denoting where the text has been changed (uploaded as a "Revised Article with Changes Highlighted" file); the other a clean version (uploaded as the article file).

(3) If available, a striking still image (a new image if one is available or an existing one from within your manuscript). If your manuscript is accepted for publication, this image may be featured on our website. Images should ideally be high resolution, eye-catching, single panel images; where one is available, please use 'add file' at the time of resubmission and select 'striking image' as the file type. 

Please provide a short caption, including credits, uploaded as a separate "Other" file. If your image is from someone other than yourself, please ensure that the artist has read and agreed to the terms and conditions of the Creative Commons Attribution License at http://journals.plos.org/plosntds/s/content-license (NOTE: we cannot publish copyrighted images). 

(4) If applicable, we encourage you to add a list of accession numbers/ID numbers for genes and proteins mentioned in the text (these should be listed as a paragraph at the end of the manuscript). You can supply accession numbers for any database, so long as the database is publicly accessible and stable. Examples include LocusLink and SwissProt.

(5) To enhance the reproducibility of your results, we recommend that you deposit your laboratory protocols in protocols.io, where a protocol can be assigned its own identifier (DOI) such that it can be cited independently in the future. For instructions see http://journals.plos.org/plosntds/s/submission-guidelines#loc-methods

While revising your submission, please upload your figure files to the Preflight Analysis and Conversion Engine (PACE) digital diagnostic tool, https://pacev2.apexcovantage.com/ PACE helps ensure that figures meet PLOS requirements. To use PACE, you must first register as a user. Then, login and navigate to the UPLOAD tab, where you will find detailed instructions on how to use the tool. If you encounter any issues or have any questions when using PACE, please email us at figures@plos.org.

We hope to receive your revised manuscript by Jan 25 2020 11:59PM. If you anticipate any delay in its return, we ask that you let us know the expected resubmission date by replying to this email.

To submit a revision, go to https://www.editorialmanager.com/pntd/ and log in as an Author. You will see a menu item call Submission Needing Revision. You will find your submission record there. 

Sincerely,

Uriel Koziol

Guest Editor

Adriano Casulli

Deputy Editor

All reviewers were generally positive regarding the manuscript and have stated that it represents a valuable contribution to the field. However, all of the reviewers expressed concerns regarding specific aspects of the manuscript and the experimental work. In particular, one main issue that two reviewers found was that only one case of E. multilocularis was investigated, and recommend either incorporating additional cases or removing this case from the manuscript (and from the title). In any case, this aspect needs to be addressed appropriately in the discussion. All reviewers also consider that the discussion should include a guideline on how cfDNA could (our could not) be used clinically in the near future as a method for diagnostics. Additional concerns should also be addressed, particularly regarding the negative controls, which should be described and presented in far greater detail. Finally, make sure to include the permit number for ethical approval of this study in the ethics statement, and to make the original data for the negative controls available as well.

Reviewer's Responses to Questions

**Key Review Criteria Required for Acceptance?**

**Methods**

-Are the objectives of the study clearly articulated with a clear testable hypothesis stated?

-Is the study design appropriate to address the stated objectives?

-Is the population clearly described and appropriate for the hypothesis being tested?

-Is the sample size sufficient to ensure adequate power to address the hypothesis being tested?

-Were correct statistical analysis used to support conclusions?

-Are there concerns about ethical or regulatory requirements being met?

Reviewer #1: see below

Reviewer #2: The objectives are clearly stated. The study design is appropriate but there are some negative control results that should be presented and discussed (these dataset is mention but missing in the text). The sample size is ok. No concerns about ethical or regulatory requirements.

Reviewer #3: The objectives of this paper were clearly described and study design is appropriate

As it could be considered as a pilot/exploratory study, number of included patient is correct

My main concerns with this study are the following: 

- among 23 patients, they had 22 CE and only one AE patient. They were included based on US examination. To my knowledge, US patterns of AE and CE are mostly different, except some cases (CL stage). In 22 out of their 23 cases, diagnosis of CE was known before biological diagnosis (molecular study)

As diagnosis of type of echinococcosis is not the key point of results but a key point of methods, i suggest that this information should be given in methods and not in results section

- the authors used surprisingly two different strategies for DNA extraction : why ? which sample among the 23 was processed with alternative approach ? 

If they processed the AE blood sample with another ADN extraction approach than that used for CE blood sample, it affect significantly all the results obtained in this study. This information is not given in the paper

- preanalytical conditions are crucial for cfDNA : the authors should described which was the delay observed between blood sampling and freezing, and centrifugation conditions (one step ? two step ? temperature of centrifugation ? speed of centrifugation g ?)

- at least, please indicate if all patients were included at diagnosis and systematically sampled before surgical and/or medical treatment and/or after beginning of treatment

**Results**

-Does the analysis presented match the analysis plan?

-Are the results clearly and completely presented?

-Are the figures (Tables, Images) of sufficient quality for clarity?

Reviewer #1: see below

Reviewer #2: Some negative control results are mention but need to be presented and discussed. The figures 3 and 4 need to be improved. Some statistical analysis/simulations in relation to the number of reads generated (more than 80 millions) and the number of reads detected may give some more support to presented results.

Reviewer #3: This study give interesting but very contrasting results between AE and CE patients, especially cfDNA yield and structure observed between the two species of Echinococcus

Except my previous limitations given previously, results were clearly expressed

**Conclusions**

-Are the conclusions supported by the data presented?

-Are the limitations of analysis clearly described?

-Do the authors discuss how these data can be helpful to advance our understanding of the topic under study?

-Is public health relevance addressed?

Reviewer #1: see below

Reviewer #2: The identification of Echinoccoccus spp. cell free DNA reads is really marginal in some patients, however the authors conclude that the technique es valid in 100% cases. This issue should be discussed and mention in conclusions.

Reviewer #3: I disagree with author's statement (Lines 419-421): if puncture is prohibited for CE patients, it is of high diagnostic value and safe in AE patients. Please see 

- Review in Clin Microbiol Review 2019 (Wen, Vuitton... your ref number one)

- Efficacy of ultrasound-guided core-needle biopsy in the diagnosis of hepatic alveolar echinococcosis: a retrospective analysis. Bulakci et al. Parasite 2016;23:19 (to be included in your references)

Include reference of crucial usefulness of imaging technique in diagnosis of echinococcosis 

Last sentence of the paper is very interesting: "given the high sequencing depth and cost, the application scenarios and potential need to be further explored". Please develop what you mean

Please balance more your results (even if very interesting contrast) : you only included one AE patient !! (Lines 411-412)

**Editorial and Data Presentation Modifications?**

Reviewer #1: see below

Reviewer #2: (No Response)

Reviewer #3: Introduction

Please include some sentences about very young cysts, called CL (please refer to review of Wen, Vuitton and to Engler et al. Simple liver cysts and cystoid lesions in hepatic alveolar echinococcosis: a retrospective cohort study with Hounsfield analysis. Parasite 2019;26 :54)

Line 396 : Baraquin (instead of Baraquin)

**Summary and General Comments**

Reviewer #1: Ji et al. report on work concerning the detection of cell-free parasite DNA in serum samples of echinococcosis patients by Next Generation Sequencing approaches. The authors identified parasite DNA in 23 samples (of 23 different patients). They demonstrated that parasite cfDNA concentrations are very low in echinococcosis patients, which could explain why several PCR approaches towards this aim so far failed (or only showed very low sensitivity). To my knowledge, successful detection of cfDNA in echinococcosis patients by high throughput NGS has so far not been published which is, indeed, a strong point of this work. I agree with the authors that basic knowledge on cfDNA realeased by these parasites into host serum is important for future development of respective molecular diagnostics methodology (although, of course, not feasible at the moment due to high costs). In general the work is well written and the NGS methods seem to have been carried out in a professional way (although a bioinformatics expert should closer evaluate this part of the work). There are, however, also several limitations in this work. On the one hand, the authors do not provide any suggestions on how, based on their data, proper molecular diagnostics of serum samples can be carried out in the future (their data rather suggest that this will not be possible for long time – which, of course, would be a valid result if properly discussed). The most serious limitation I see is that only one E. multilocularis sample was analyzed. This is surely not enough to come to significant conclusions, particularly concerning the general suitability of echinococcosis serum samples for molcular diagnostics (in E. multilocularis the cfDNA concentration was much higher), or concerning differential diagnostic methods that can distinguish between E. granulosus and E. multilcularis. On the one hand, this could be solved by taking out the E. multilocularis sample and by concentrating only on E. granulosus (stating, in the end, that it is almost impossible to detect E. granulosus DNA in plasma samples). On the other hand, I would consider a significant step forward (i.e. ‚importance‘ as outlined in the PNTD guidelines for publication) in echinococcosis diagnostics the comprehensive analysis of both E. multilocularis and E. granulosus cfDNA in patient plasma. I thus strongly suggest that additional E. multilocularis samples are analyzed so that the title of the study (Comprehensive characterization of cf Echinococcus spp. DNA….) is justified. 

Additional points

1) The English of the manuscript is generally quite well and comprehensible. Nevertheless, another round of checking readability would be good (particularly proper use of specifying articles like ‚the‘ and ‚a‘ as well as correct use of plural/singular).

2) L146-148: why is it likely that parasite DNA can be found in plasma because of complex host-parasite interaction? This kind of interaction could exclusively be mediated by proteins.

3) Shouldn’t there be a permit no. for the ethics statement?

4) Paragraph 2.2.: please be specific on when the plasma samples have been taken. I guess it was before (!) treatment. This should be pointed out in detail.

5) Paragraph 2.5.: please be specific on when the ELISA has been carried out. 

6) Related to point 4 and 5: it would, of course, strengthen the study if cfDNA analyses (and maybe ELISA) would have been carried out also after treatment (especially chemotherapy) at least for some samples.

7) The authors only superficially discussed why they detected so much more cfDNA in the E. multilocularis patient. I think the big difference is that E. multilocularis metacestode material in the human host is generally a mixture between nectrotic parasite tissue and actively proliferating tissue. Hence, cfDNA could result from parasite cell necrosis. This should be taken into account and could also explain why there were shorter fragments in the E. multilocularis sample.

8) L 437/438: shouldn’t it be possible to calculate reads per genome size? This would give us hints as to whether nuclear or mitochondrial genome are over-represented.

9) Fig. 3: At least in the mitochondrial genome I see some hotspots. Which are these?

10) Table 1: please add for all samples which species was identified according to NGS reads. 

11) References: genus/species in italics.

Reviewer #2: Review of “Comprehensive characterization of cell-free Echinococcus spp. DNA in echinococcosis patients’ plasma using extremely high throughput sequencing” by Ji and co-authors.

In the present manuscript (MS) Ji and coauthors report the results of applying extremely high throughput sequencing to detecting cell-free Echinococcus spp. DNA in plasma of echinococcosis patients.

Authors do a well-organizer presentation of auspicious results regarding the possibility of identified and partially characterize Echinococcus spp. infections from human blood samples. The MS followed a straight-forward bioinformatic pipeline to identify reads from cell free DNA from Echinococcus and more importantly to discard false positive results. This approaches has been successfully applied for others pathogens.

This MS is well written and my overall impression of it is positive, however I have some concerns that I think the authors need to address before publications. Some suggested changes could improve the robustness and presentation of the MS.

Major Comments:

1. Negative control data are not mention (Humans with no diagnostic Echinoccoccus) until the Results sections. This is a major problem since no results about this control dataset is presented in the MS (see Line 295, pg. 8). Given that detection of Echinoccoccus spp. reads is so marginal in the whole NGS dataset, the results of the bioinformatics pipeline applied to this 107 human blood samples are key to support presented positive results. Negative controls are not included under accession number of CNGB.

2. The number of reads identified is rather marginal; this is particularly noticeable when considering the millions of reads generated in each sequencing experiment (median 235 million reads). Authors should discuss more thoroughly these results and possible improvements that need to be done in order to make this approach clinically feasible. For instance, I wonder if single reads could improve the detection of cell free DNA. In relation to this the authors should consider do some statistical/modeling analysis to estimate the minimal number of reads generated to get positive results.

Minor Comment (not in order):

• When building the Echinococcus spp. reference database, in order to reduce

sequence contamination and get high quality genomes, authors used relatively high thresholds values (97% identity, 92% coverage, e-value) when filtering the posible sequences from host (human, mouse, etc..). Reads with high similarity to the host genomes are discarded. Authors need to clarify why these values are chosen, and as I understand the higher thresholds may allow including contaminant reads in the analysis.

• I am not quite sure if Kraken is the best option to do the “mapping”. Please check or clarify.

• More analysis on the distribution of identified reads in the genome of E. spp. are needed. Are there any difference among patients? Which are the more represented regions? Is there any common characteristic in the overrepresented regions? 

• Please check figures 3 and 4, both are not in resolution and are not particularly informative about genome distribution of reads.

• Clarify the meaning of CL and CE in Table 1.

Reviewer #3: As a summary, very interesting paper and well-done study. 

Authors should improve description of methods

If they processed blood from their AE patient with a distinct extraction, it is a bias. Thus, they should 

- either clearly balance their results and discussion

- or limit their paper to their 22 CE patients (they may design another similar study focused on AE patients)

PLOS authors have the option to publish the peer review history of their article (what does this mean?). If published, this will include your full peer review and any attached files.

Reviewer #1: No

Reviewer #2: No

Reviewer #3: No

---

## [Decision Letter · Decision Letter 1]

18 Feb 2020

Dear Mr Ji,

We are pleased to inform you that your manuscript 'Comprehensive characterization of plasma cell-free Echinococcus spp. DNA in echinococcosis patients using extremely high-throughput sequencing.' has been provisionally accepted for publication in PLOS Neglected Tropical Diseases.

Before your manuscript can be formally accepted you will need to complete some formatting changes, which you will receive in a follow up email. A member of our team will be in touch within two working days with a set of requests.

Best regards,

Uriel Koziol

Guest Editor

Adriano Casulli

Deputy Editor

The authors have succesfully reviewed their manuscript, addressing all of the main concerns raised by the reviewers.

Reviewer's Responses to Questions

**Key Review Criteria Required for Acceptance?**

**Methods**

-Are the objectives of the study clearly articulated with a clear testable hypothesis stated?

-Is the study design appropriate to address the stated objectives?

-Is the population clearly described and appropriate for the hypothesis being tested?

-Is the sample size sufficient to ensure adequate power to address the hypothesis being tested?

-Were correct statistical analysis used to support conclusions?

-Are there concerns about ethical or regulatory requirements being met?

Reviewer #1: To my opinion the authors have adequately addressed not only my concerns on the first version but also the concerns of all other reviewers. They have made additional analyses, provided a more detailed description of the methods, and toned down several conclusions when appropriate. The revised version of the manuscript is a well-perormed improvement of the first version and the scientific discussion in the response to reviews was thorough and inspiring. Congratulations.

Reviewer #2: The authors take in account almost all our comments.

Reviewer #3: (No Response)

**Results**

-Does the analysis presented match the analysis plan?

-Are the results clearly and completely presented?

-Are the figures (Tables, Images) of sufficient quality for clarity?

Reviewer #1: see above

Reviewer #2: The authors modified figures 3 and 4 as suggested, did statistical analyses, clarified specific sections of the MS and response to all comments raised by reviewers.

Reviewer #3: (No Response)

**Conclusions**

-Are the conclusions supported by the data presented?

-Are the limitations of analysis clearly described?

-Do the authors discuss how these data can be helpful to advance our understanding of the topic under study?

-Is public health relevance addressed?

Reviewer #1: see above

Reviewer #2: (No Response)

Reviewer #3: (No Response)

**Editorial and Data Presentation Modifications?**

Reviewer #1: I did not find any obvious errors.

Reviewer #2: Accept

Reviewer #3: (No Response)

**Summary and General Comments**

Reviewer #1: see above and first review

Reviewer #2: Authors have addressed all my comments into this new version of the manuscript. In general I consider this is a well-presented paper with a significant impact on the specific field. Now, I believe this MS is now suitable for publication in PlosNTD.

Reviewer #3: The authors have clearly improved the quality of the manuscript. They have qualified their comments about the EA patient. All my previous concerns were solved. The authors provided us a clear and innovative manuscript. Thank you.

PLOS authors have the option to publish the peer review history of their article (what does this mean?). If published, this will include your full peer review and any attached files.

Reviewer #1: Yes: Klaus Brehm

Reviewer #2: No

Reviewer #3: No

---

## [Editor Report · Acceptance letter]

2 Apr 2020

Dear Mr Ji,

We are delighted to inform you that your manuscript, "Comprehensive characterization of plasma cell-free *Echinococcus* spp. DNA in echinococcosis patients using ultra-high-throughput sequencing.," has been formally accepted for publication in PLOS Neglected Tropical Diseases.

Best regards,

Serap Aksoy

Editor-in-Chief

Shaden Kamhawi

Editor-in-Chief
